# DNA translocation mechanism of an XPD family helicase

Kaiying Cheng, Dale B Wigley*

Section of Structural Biology, Department of Medicine, Imperial College London, London, United Kingdom

**Abstract** The XPD family of helicases, that includes human disease-related FANCJ, DDX11 and RTEL1, are Superfamily two helicases that contain an iron-sulphur cluster domain, translocate on ssDNA in a 5′−3′ direction and play important roles in genome stability. Consequently, mutations in several of these family members in eukaryotes cause human diseases. Family members in bacteria, such as the DinG helicase from *Escherichia coli*, are also involved in DNA repair. Here we present crystal structures of complexes of DinG bound to single-stranded DNA (ssDNA) in the presence and absence of an ATP analogue (ADP•BeF$_3$), that suggest a mechanism for 5′−3′ translocation along the ssDNA substrate. This proposed mechanism has implications for how those enzymes of the XPD family that recognise bulky DNA lesions might stall at these as the first step in initiating DNA repair. Biochemical data reveal roles for conserved residues that are mutated in human diseases.

DOI: https://doi.org/10.7554/eLife.42400.001

## Introduction

DNA helicases are enzymes that separate nucleic acid duplexes into their component strands (*Singleton et al., 2007*). They have been divided into a number of superfamilies on the basis of sequence similarities (*Gorbalenya and Koonin, 1993*) that have also proved to reveal similarities in mechanism. Although most family members unwind nucleic acid duplexes some, such as chromatin remodellers (*Clapier et al., 2017*), instead translocate along nucleic acid duplexes without unwinding them and others translocate along single-stranded substrates to displace bound proteins (*Jankowsky et al., 2001*). The two largest helicase/translocase families are Superfamily 1 (SF1) and Superfamily 2 (SF2). The latter superfamily contains the canonical 'DEAD' box RNA helicases – a large group of proteins with roles in unwinding RNA substrates in processes such as transcription, splicing and ribosome assembly (*Bourgeois et al., 2016*). However, this family also contains DNA helicases with roles in a variety of different biological processes.

SF1 and SF2 members are generally monomeric enzymes with ATP-dependent translocation on single-stranded DNA (ssDNA) templates that has a characteristic polarity of either 5′−3′ or 3′−5′ (*Singleton et al., 2007*). However, many of these enzymes interact with other proteins in larger, multi subunit complexes (such as RecBCD that contains two SF1 helicases with opposite polarities (*Wigley, 2013*). Others clearly act cooperatively (*Eoff and Raney, 2010*). The crystal structures of several proteins have revealed how these enzymes interact with ssDNA and some studies have obtained multiple catalytic states of the proteins that have allowed deduction of their translocation mechanisms. The first enzyme for which this was achieved was the SF1 3′−5′ helicase PcrA (*Subramanya et al., 1996*; *Velankar et al., 1999*). Similar nucleotide-bound states of the SF2 3′−5′ helicase NS3 also revealed how this enzyme couples ATP hydrolysis to ssDNA (or ssRNA) translocation (*Gu and Rice, 2010*) by a mechanism that is distinct from SF1 enzymes like PcrA. For 5′−3′ translocation, structures of the SF1 enzyme RecD2 were also determined to provide yet another translocation mechanism (*Saikrishnan et al., 2009*). In all three of these example enzymes, although

*For correspondence:
d.wigley@imperial.ac.uk

Competing interests: The authors declare that no competing interests exist.

the mechanisms are distinct, the translocation step size that is deduced is one ATP per base. However, an obvious gap in our mechanistic understanding of SF1 and SF2 enzymes is that of a SF2 enzyme capable of 5'−3' translocation.

XPD family enzymes are SF2 helicases with 5'−3' directionality and some, such as XPD itself, recognise bulky DNA lesions (*Voloshin et al., 2003*; *Wirth et al., 2016*). XPD itself has a role in DNA repair in archaea and eukaryotes (*Houten et al., 2016*), although it is not clear that the functions are equivalent in these different kingdoms of life. XPD is a part of the TFIIH complex that does not have a direct equivalent in eubacteria although *Escherichia coli* contains homologues of XPD (DinG) and XPB (RadD), both of which are subunits of TFIIH in eukaryotes. DinG and RadD are implicated in DNA repair (*Voloshin et al., 2003*; *Thakur et al., 2014*; *Chen et al., 2015*) although their precise functions are poorly defined. Other eukaryotic XPD family members (e.g. RTEL1, FANCJ, DDX11) also have roles in DNA repair as well as other cellular processes including recovery of stalled replication forks and bypassing DNA lesions (*Vannier et al., 2014*; *Bharti et al., 2016*; *Abe et al., 2018*). Consequently, mutations in these genes are linked to cancer and other human diseases (*Cerami et al., 2012*; *Suhasini and Brosh, 2013*; *Guo et al., 2016*; *Krassowski et al., 2018*). XPD is the only family member for which there is any structural information and this has been limited to crystal structures (*Liu et al., 2008*; *Fan et al., 2008*; *Wolski et al., 2008*; *Kuper et al., 2012*; *Constantinescu-Aruxandei et al., 2016*) or cryoEM at medium resolution of the protein in the context of TFIIH (*Greber et al., 2017*; *Schilbach et al., 2017*). None is visualised with a ssDNA substrate contacting the entire ssDNA-binding site, or with any ATP analogue. Consequently, a molecular understanding of the mechanism of XPD family enzymes is currently lacking.

DinG is another member of the XPD family, that appears to have a poorly defined role in DNA repair (*Voloshin et al., 2003*). Additionally, it appears to have a role, by cooperation with UvrD and Rep helicases, both in removing R-loops and removing with RNA polymerase at blocked replication forks (*Boubakri et al., 2010*). DinG shows specificity for a number of unusual DNA structures such as G-quadruplexes as well as D-loops (*Voloshin et al., 2003*; *Thakur et al., 2014*; *Voloshin and Camerini-Otero, 2007*), a property that is also shared by some other XPD family members like FANCJ (*Bharti et al., 2016*). DinG has also been reported to have a role in a type I CRISPR system in *Acidithiobacillus ferrooxidans* (*Makarova et al., 2011*). These CRISPR-associated DinG proteins are unusual because they lack FeS clusters and contain an N-terminal extension that may be a nuclease domain (*McRobbie et al., 2012*). The nuclease activity is likely to play an important role in spacer acquisition since these CRISPR systems appear to lack Cas1 and Cas2 proteins.

Despite intensive effort, high resolution structural information about the molecular mechanism of XPD family enzymes has remained elusive. Here, we present the crystal structures of two complexes of *E. coli* DinG; a binary complex bound to a ssDNA substrate, and a ternary complex with ssDNA and an ATP analogue (ADP•BeF$_3$), that provide information about how these enzymes translocate on DNA. As observed in crystal structures of other SF1 and SF2 helicases (*Velankar et al., 1999*; *Gu and Rice, 2010*; *Saikrishnan et al., 2009*; *Sengoku et al., 2006*), binding of nucleotide induces a conformational change that involves closure of a cleft between the two canonical helicase domains (*Singleton et al., 2007*). However, additional changes occur in the detailed interactions between the protein and the bound ssDNA that result in unidirectional translocation in a 5'−3' direction by a mechanism that is supported by biochemical data. The structures also suggest a potential mechanism for how some members of the XPD family of enzymes might recognise and stall at bulky lesions. The structures represent an important advance in our understanding of the mechanism of XPD family enzymes as well as explaining how some of the mutations in these proteins cause deficiencies in their activities.

## Results and discussion

### Overall structure of DinG and comparison with XPD

DinG shares significant similarity to XPD at both the structural and sequence level (*Liu et al., 2008*; *Fan et al., 2008*; *Wolski et al., 2008*; *Kuper et al., 2012*; *Constantinescu-Aruxandei et al., 2016*) (*Figure 1*, *Figure 1—figure supplement 1*, *Figure 1—figure supplement 2*). The proteins comprise four domains; the helicase (HD1 and HD2), arch and FeS cluster domains (*Figure 1*, *Figure 1—figure supplement 1* and *Figure 1—figure supplement 2*). The mean r.m.s. deviation of Cα positions for

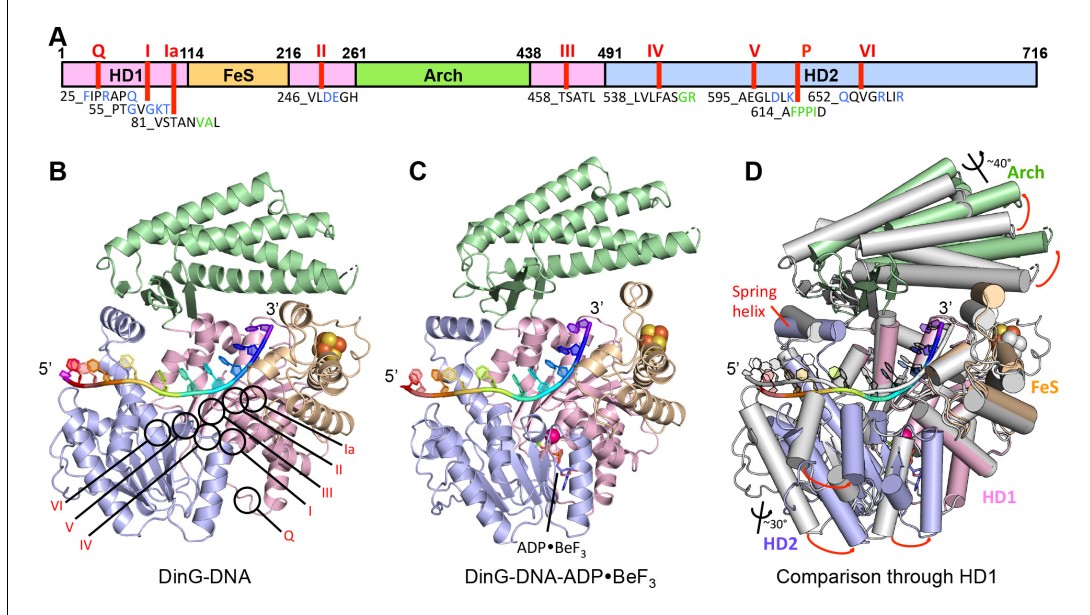

**Figure 1.** Structures of the DinG/ssDNA binary and ternary complexes. (**A**) The domain arrangement of DinG. The HD1, HD2, 4FeS and Arch domains are coloured in pink, lilac, wheat and green, respectively. Conserved motifs are labelled and listed. Residues labelled in blue participate in ATP hydrolysis, and those in green participate in DNA binding. (**B**) Cartoon view of the DinG binary complex with ssDNA coloured as in (**A**). (**C**) Cartoon view of the DinG ternary complex with ssDNA and ADP•BeF3, coloured as in (**A**). (**D**) Superposition of the two structures based on the HD1 domain to illustrate the conformational changes between the structures. The binary complex is in grey and the ternary complex is coloured by domains as in (**A**).
DOI: https://doi.org/10.7554/eLife.42400.002

The following figure supplements are available for figure 1:

**Figure supplement 1.** Sequence and structural alignment of DinG/XPD family members.
DOI: https://doi.org/10.7554/eLife.42400.003

**Figure supplement 2.** Superimposition of DinG binary structure and human XPD structure from the TFIIH complex.
DOI: https://doi.org/10.7554/eLife.42400.004

**Figure supplement 3.** The FeS (iron-sulphur cluster) domain.
DOI: https://doi.org/10.7554/eLife.42400.005

each of the helicase and arch domains of DinG, superimposed individually on equivalent domains from XPD structures, ranges from 1.8 to 3.3 Å, comparable to the same comparisons between different XPD crystal structures. The canonical SF2 helicase domains (HD1 and HD2) have the expected fold seen in other structures (*Singleton et al., 2007*), with an ATP-binding site located between them. The FeS domain, that is inserted within the HD1 sequence, contains an iron-sulphur (FeS) cluster at its centre (*Figure 1* and *Figure 1—figure supplement 3*). Although the FeS domains in XPD and DinG are both all α-helical, are topologically related with respect to coordination of the FeS cluster and occupy very similar positions in the structure, they differ somewhat in fold and sequence (*Figure 1—figure supplement 3*). The arch domain sits above the other domains and contacts the FeS domain above the ssDNA binding surface. However, the similarity of the helicase and arch domains (that together comprise >85% of the protein), to those in XPD, suggest that DinG is a good structural model for other XPD family enzymes.

## Structure of the DinG binary complex with ssDNA

The DinG binary complex structure was solved at 2.5 Å resolution by two wavelength anomalous diffraction data based on selenomethionine-substituted protein but the final model was built and refined using data from a crystal of native protein. Crystallographic statistics are given in *Table 1*. Although a twelve base oligo-dT substrate was used in the crystallisation, only eleven bases are evident in the structure suggesting these likely comprise the entire ssDNA-binding site (*Figure 2* and *Figure 2—figure supplement 1*).

**Table 1.** Statistics from crystallographic analysis.

| Complex | Se-DinG-DNA | | DinG-DNA | DinG-DNA-ADPBeF$_3$ |
|---|---|---|---|---|
| PDB ID | - | - | 6FWR | 6FWS |
| Data collection | | | | |
| Source | I04 | | I04 | I04-1 |
| Wavelength (Å) | 0.9795 (peak) | 0.9860 (remote) | 0.9795 | 0.9159 |
| Resolution (Å) | 59–2.6 (2.64–2.6) | 56.2–3.4 (3.46–3.4) | 79.59–2.5 (2.54–2.5) | 63.37–2.5 (2.54–2.5) |
| Space group | P2$_1$2$_1$2 | P2$_1$2$_1$2 | P2$_1$2$_1$2 | P2$_1$2$_1$2$_1$ |
| Cell dimensions: a, b, c | 99.11, 134.47, 59 | 99.11, 134.47, 59 | 98.97, 133.87, 58.86 | 109.77, 119.75, 126.74 |
| Obeservation | 274170 (12271) | 124833 (6193) | 166828 (8129) | 266197 (13044) |
| Unique reflections | 24842 (1169) | 11591 (559) | 27857 (1351) | 58152 (2827) |
| R$_{merge}$ (%) | 12.3 (51.7) | 15.5 (43.7) | 10.9 (50.3) | 11.9 (54.1) |
| I/σI | 25.8 (4.7) | 13.7 (4.9) | 12.3 (3.3) | 9.6 (2.6) |
| Completeness (%) | 99.2 (95.3) | 99.5 (96.7) | 100 (98) | 99.5 (98) |
| Redundancy | 11 | 10.8 | 6.0 | 4.6 |
| Refinement statistics | | | | |
| Resolution (Å) | - | - | 79.95–2.5 (2.54–2.5) | 63.37–2.5 (2.54–2.5) |
| R$_{factor}$ (%)/R$_{free}$ (%) | - | - | 21.08/24.57 | 21.91/25.57 |
| rmsd bonds (Å)/angles (°) | - | - | 0.008/1.134 | 0.008/1.080 |
| Ramachandran plot: Favored (%) | - | - | 97.4 | 96.2 |

The numbers in parentheses refer to the last shell.

R$_{factor}$ = $\Sigma$||F(obs)- F(calc)||/$\Sigma$|F(obs)|.

R$_{free}$ = R factor calculated using 5.0% of the reflection data randomly chosen and omitted from the start of refinement.

DOI: https://doi.org/10.7554/eLife.42400.012

Overall, the DNA is kinked but with many bases stacked upon one another along the strand in a conformation that is quite different to that observed previously in any other SF1 or SF2 protein (*Figure 2—figure supplement 5*). However, the bases contacting HD1 are similar to those in an equivalent position in NS3 except that in NS3 there is a tryptophan residue intercalated between the bases (*Gu and Rice, 2010*; *Kim et al., 1998*) that is not conserved in DinG. The bases spanning the HD1 domain, and crossing over to the HD2 domain, are stacked on one another until they encounter a proline-rich strand (P motif) on the HD2 domain, at which point the next base would clash with this strand if stacked on the preceding one so it is flipped up and over this strand. In all, three bases are contacted by the HD1 domain and seven bases contact the HD2 domain, with an additional base spanning the gap between the domains. Two structures of XPD proteins also have bound ssDNA (*Kuper et al., 2012*; *Constantinescu-Aruxandei et al., 2016*). However, despite being longer oligonucleotides, only four or five bases are ordered in these structures and they only make contacts with the HD2 domain. Furthermore, the details of the interactions in each structure are inconsistent and both structures differ from those we observe in DinG (*Figure 2—figure supplement 3*). The reasons for this are unclear although in one case the DNA was chemically crosslinked to the protein, which may have induced some distortions.

The bound ssDNA sits in a positively-charged groove across the ATPase domains in a similar location, and with the same directional polarity, seen in other SF1 and SF2 helicases (*Singleton et al., 2007*), and the contacts are mainly with the phosphodiester backbone (*Figure 2*). Although largely polar hydrogen bonding interactions, some exceptions include two hydrophobic stacking interactions (F615 and F638) with deoxyribose moieties that help to discriminate between DNA and RNA substrates (*Figures 2* and *4*), as observed previously in other helicases (*Saikrishnan et al., 2009*).

In addition to the DNA backbone contacts there are a few contacts with the DNA bases (*Figure 2*). Notably, there is an aromatic stacking interaction with Tyr636 intercalated between the first and second bases at the 5' end of the ssDNA on the HD2 domain. Although a similar base intercalation interaction is seen in NS3 (*Kim et al., 1998*; *Gu and Rice, 2010*), this is at the opposite end of

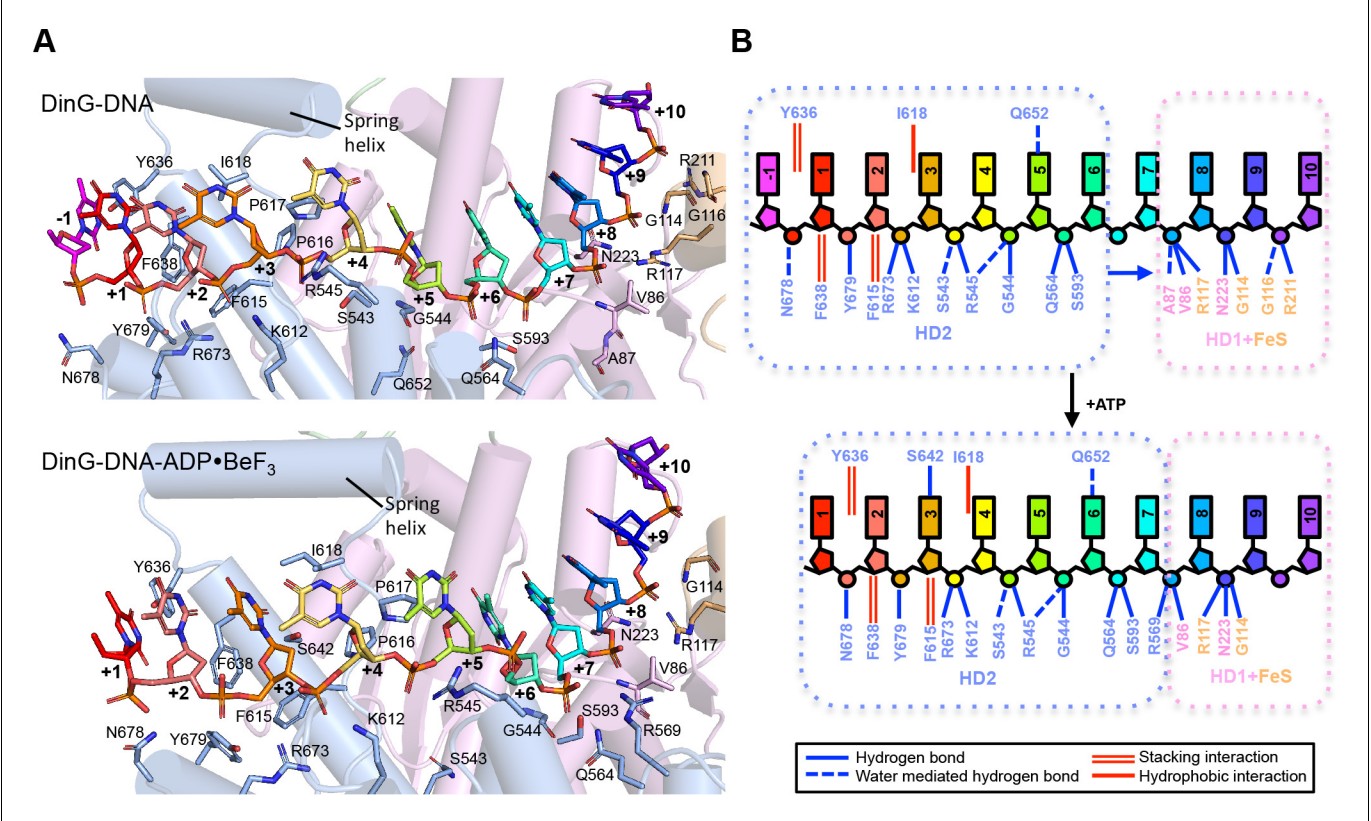

**Figure 2.** Details of the ssDNA-binding site. (**A**) The DNA-binding site for the binary (above) and ternary (below) complexes. The side chains of residues participating in DNA binding are shown as sticks, coloured according to the domains from which they originate (HD1 in pink, HD2 in pale blue). The nucleotides of the bound ssDNA are rainbow coloured in the 5' (red) to 3' (blue) direction. The colour scheme for the bases is the same in all figures. (**B**) Cartoon representation of the same views detailing the contacts between the protein and DNA in each complex.

DOI: https://doi.org/10.7554/eLife.42400.006

The following figure supplements are available for figure 2:

**Figure supplement 1.** Electron density corresponding to bound ssDNA.
DOI: https://doi.org/10.7554/eLife.42400.007
**Figure supplement 2.** Electron density corresponding to bound nucleotide.
DOI: https://doi.org/10.7554/eLife.42400.008
**Figure supplement 3.** Comparison of 5' DNA-binding in DinG and XPD.
DOI: https://doi.org/10.7554/eLife.42400.009
**Figure supplement 4.** DNA binding sites on DinG and XPD FeS domain.
DOI: https://doi.org/10.7554/eLife.42400.010
**Figure supplement 5.** ssDNA conformation in SF1 and SF2 helicases.
DOI: https://doi.org/10.7554/eLife.42400.011

the bound DNA at the 3' end and involves a tryptophan residue from the HD1 domain. The bases spanning the HD1 domain, and crossing over to the HD2 domain, are stacked on one another until they encounter the P motif (residues 614-AFPPID-619). The P motif is located in front of the spring-helix and contains several hydrophobic residues, often prolines, that would rigidify this region of the structure. The P motif in the taXPD structure (*Kuper et al., 2012*; *Constantinescu-Aruxandei et al., 2016*), also forms a narrow channel (*Figure 2—figure supplement 3*). The major protein-DNA interactions at the HD1/FeS domain interface are mediated by two arginine residues, R117 and R211 (*Figure 1—figure supplement 3*). Although the structures of the FeS domain differs between XPD and DinG, there are two basic residues (R88 and K170 in taXPD) that are conserved and interact with a sulphate ion that mimics the DNA phosphodiester backbone (*Figure 1—figure supplement 3* and *Figure 2—figure supplement 4*).

## Structure of DinG ternary complex with ssDNA and ADP•BeF₃

The DinG ternary complex structure was solved at 2.5 Å resolution by molecular replacement using individual domains of the binary structure, and crystallographic statistics are given in *Table 1*. The domain architecture of the ternary complex is similar to the binary complex, although there are changes in the relative orientations of the domains (*Figure 1*). Due to these changes, the protein now interacts with only ten bases of ssDNA (*Figure 2*). Although there are still three bases contacting HD1 and seven contacting HD2, there is no longer a base spanning the gap between the domains indicating that one of the domains must have slid along the ssDNA onto this base. Since the enzyme has 5′−3′ directionality (*Voloshin et al., 2003*), this would suggest it is the HD2 domain that has done the sliding. Consistent with this notion, the base at position +5 (*Figure 2*) no longer stacks on base +6 (as seen in the binary complex) but instead would now clash with the P motif so becomes flipped up and over the strand to occupy the position taken by the +4 base in the binary complex. Similarly, the contacts with other bases spanning the HD2 domain are displaced by one base compared to the binary complex. Although many of the ssDNA contacts are similar in both complexes, there are additional contacts between the HD1/FeS domains and the ssDNA in the binary complex than in the ternary complex (*Figure 2*). By contrast, the number of contacts between ssDNA and the HD2 domain is greater in the ternary complex. The implications of these differences between the structures are discussed below in regard to the translocation mechanism. The principal difference between the structures of DinG in the presence and absence of ADP•BeF₃ is a conformational change (*Figure 1*) similar to that observed for other SF1 and SF2 helicase family members (*Singleton et al., 2007*). As expected, there is an ADP•BeF₃ moiety bound at the interface between the HD1 and HD2 domains (*Figure 2—figure supplement 2* and the nucleotide is contacted by several residues in a manner seen previously in other helicases (*Singleton et al., 2007*). This conformational change results when the cleft between the two canonical helicase/translocase domains closes around the bound nucleotide. The consequences of this conformational change are discussed below.

## A mechanism for translocation of ssDNA in a 5′−3′ direction

Translocation on ssDNA can be in either a 3′−5′ or 5′−3′ direction (*Singleton et al., 2007*). Mechanisms for 3′−5′ translocation have been described in structural detail for both SF1 (PcrA, (*Velankar et al., 1999*)) and SF2 (NS3, (*Gu and Rice, 2010*)) helicases. A mechanism for 5′−3′ translocation for a SF1 helicase has also been reported (RecD2, (*Saikrishnan et al., 2009*)). The mechanisms all suggest a step size of one base per ATP hydrolysed, which is consistent with biochemical measurements (*Dillingham et al., 2000*).

The principal differences between the structures we present here are changes in the interactions of the HD1 and HD2 domains that result in sliding of the HD2 domain along the bound ssDNA when ATP binding induces a conformational change (*Figure 1 and 2*). These changes allow us to propose a mechanism for ssDNA translocation by SF2 5′−3′ enzymes such as the XPD family enzymes (*Figure 3* and *Video 1*). Upon initial binding to ssDNA, the protein makes interactions via a groove running across the surface of the ATPase (motor) domains. Upon ATP binding, a conformational change causes the HD2 motor domain to slide along ssDNA, while the HD1 domain retains a tight grip. When ATP is hydrolysed, the enzyme relaxes to the apo state allowing the HD1 domain to slide along ssDNA while the HD2 domain now grips tightly. As with translocation mechanisms for other SF1 and SF2 enzymes (*Velankar et al., 1999*; *Gu and Rice, 2010*; *Saikrishnan et al., 2009*), the key to the mechanism is the regulation of grip for each of the motor domains and how that changes in response to the different nucleotide-bound states of the enzyme. For DinG, in the binary complex, the HD1 domain has a tighter grip than in the ATP-bound (ternary) state. This is achieved by increasing the number of contacts with the residues towards the 3′-end of the ssDNA binding site (*Figure 2*). By gripping the ssDNA more tightly than the HD2 motor domain, as the conformational change induced by ATP binding closes the cleft between the motor domains, this causes the ssDNA to slide across the surface of the HD2 domain. In doing so, there is a steric clash between the base at position +5 and the rigid P motif that forces the base out of a stacked conformation with the adjacent base. After ATP hydrolysis, changes occur that cause HD2 to now have a tighter grip on the 5′ bases (again due to an increased number of interactions) and thus, as the conformational changes open the cleft between the motor domains, the ssDNA slides across the HD1 domain. The

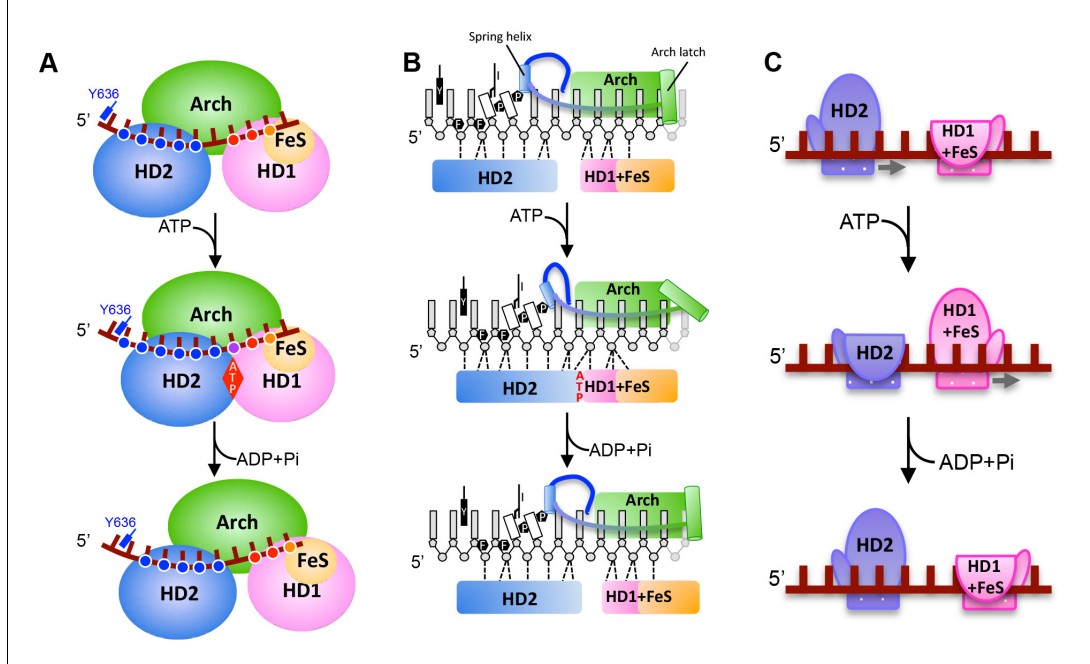

**Figure 3.** A proposed mechanism for 5′−3′ translocation. Cartoons depicting the conformational changes and events during DNA translocation at (**A**) the domain level, (**B**) detailed interactions between the bases of the bound ssDNA with the HD1 +FeS and HD2 domains during a translocation cycle. Dotted lines shown contacts between the protein and DNA. (**C**) cartoon showing how the HD1 and HD2 domains alter grip on the bound ssDNA at different steps in the translocation. Further details are provided in the text.

DOI: https://doi.org/10.7554/eLife.42400.013

The following figure supplement is available for figure 3:

**Figure supplement 1.** Translocation mechanisms for SF1 and SF2 helicases.

DOI: https://doi.org/10.7554/eLife.42400.014

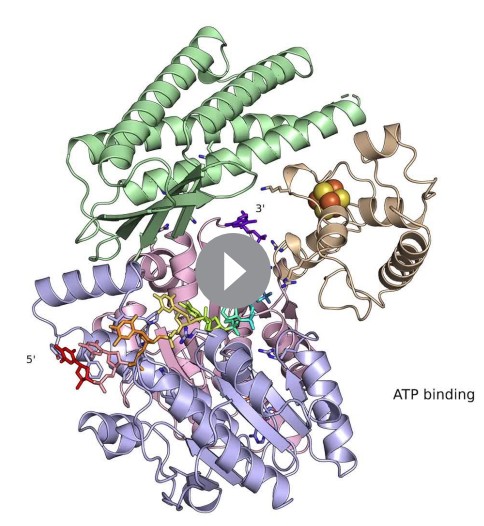

**Video 1.** Conformational changes associated with ATP binding. A morph between the two structures showing how ssDNA is translocated across the surface of the protein in response to ATP binding.

DOI: https://doi.org/10.7554/eLife.42400.015

consequence of these two steps is translocation along the bound ssDNA in a 5′−3′ direction, first across the HD2 domain as ATP binds and then across the HD1 domain after ATP hydrolysis. We note, however, that this is a simple two-step model and we cannot preclude additional steps in the overall reaction cycle but this (likely over simplified) model allows a framework to explain directional translocation by these enzymes in the same way that similar structures have provided a framework to explain translocation by other SF1 and SF2 enzymes (*Velankar et al., 1999*; *Gu and Rice, 2010*; *Saikrishnan et al., 2009*).

Although this translocation mechanism is different to that proposed for other helicases, it retains a step size of one ATP per base translocated as deduced for other SF1 and SF2 family enzymes (*Velankar et al., 1999*; *Gu and Rice, 2010*; *Saikrishnan et al., 2009*; *Qi et al., 2013*). Previous studies have revealed mechanisms for both 3′−5′ and 5′−3′ translocation in SF1 helicases but just 3′−5′ translocation by SF2 enzymes. The work we present here now completes our picture for the mechanisms of

translocation in each of the two polarities by the two main families of monomeric, single-strand translocases (*Figure 3—figure supplement 1*).

## Biochemical data in support of the model

In order to test the validity of our model, we created a number of site-directed mutant proteins and assessed their biochemical activities (*Figure 4* and *Figure 4—figure supplement 1*). The residues selected all play important roles in the interaction with nucleic acid and the translocation mechanism we propose. The mutant proteins all show defects in various aspects of their biochemical activities to differing degrees. All of the mutants showed defects in ssDNA binding although these were minor for the R117A and R211A mutations in the FeS domain (*Figure 4*). As a consequence, although we tried to compare ATPase rates, these are very low in the absence of ssDNA (*Voloshin et al., 2003*) and we were unable to saturate most of the mutants with ssDNA for the assays. As expected, the defects in interaction with ssDNA also resulted in at least 100-fold reduction in helicase activity (*Figure 4*). Helicase activity was severely impaired even for the R117A and R211A mutations that have lesser effects on ssDNA binding than the other mutations. These data confirm the importance of the residues we have highlighted in the interaction with the DNA substrate and consequently for the translocation mechanism. Consistent with the essential roles we propose, several of the corresponding residues in XPD, FANCJ, DDX11 or RTEL1 are reported to be disease-related (*Figure 4*). Indeed, mutations in these residues across the ssDNA-binding site in all three proteins are linked to human diseases (*Suhasini and Brosh, 2013*; *Guo et al., 2016*; *Krassowski et al., 2018*) and explain the molecular basis for the deficiencies in activity of these mutant proteins.

## Implications for DNA damage recognition by XPD family helicases

The similarity of DinG to other XPD family members provides broader insight into the mechanism of this important class of proteins. Structural studies of XPD (*Liu et al., 2008*; *Fan et al., 2008*; *Wolski et al., 2008*; *Kuper et al., 2012*; *Constantinescu-Aruxandei et al., 2016*) have revealed detail about the overall structure of the protein and its interaction with ssDNA. However, only nucleotide-free states have been crystallised and also the observed interactions with bound ssDNA do not span the entire binding site and are not consistent between structures. Consequently, there is a limited understanding of how XPD interacts with DNA, the mechanism of translocation, how it unwinds DNA substrates or how it recognises DNA lesions.

The step size of one base also raises an intriguing possibility with regard to recognition of bulky DNA lesions by those members of the XPD family that recognise such lesions. A central component of the translocation is the requirement for single bases to flip out of a stacked arrangement as they collide with the rigid and hydrophobic P motif. One of the major lesions recognised by XPD is a thymidine dimer in which adjacent thymidine bases in one strand become crosslinked to one another. Such an arrangement now has consecutive bases locked together so would require them both to flip together with a step of two bases. Consequently, this would be blocked from flipping during translocation and this may constitute one aspect of the damage recognition process, causing the enzyme to stall at these lesions as a signal for repair to initiate. Indeed, consistent with this hypothesis, there is a large hydrophobic pocket on the protein surface in this location, particularly in the binary complex, that could accommodate a variety of bulky lesions such as thymidine dimers (*Figures 2* and *5*). Our structures show that the pocket dimensions alter during ATP binding and hydrolysis (*Figure 5*). If the pocket were occupied with a bulky lesion then domain closure would be prevented and hence ATPase would be blocked and translocation would stall (*Figure 5—figure supplement 1*). Although further work is required to confirm the effects of these changes on enzyme stalling, it has been shown that XPD stalls at bulky lesions on the translocating strand (*Wirth et al., 2016*).

Our data here expand upon our knowledge of the ssDNA binding site and a mechanism for translocation by these enzymes, as well as supporting a potential mechanism for recognition of base lesions. The roles of different residues in the ssDNA binding site have additional implications for known disease-causing mutations in XPD and other family members. Furthermore, we have mutated several residues in the ssDNA-binding site and several of these have equivalents in XPD, FANCJ, DDX11 or RTEL1 that are associated with human diseases. We can now ascribe a function to these residues to explain the basis of the malfunction of these proteins and expand our understanding of diseases related to mutations in these XPD family members.

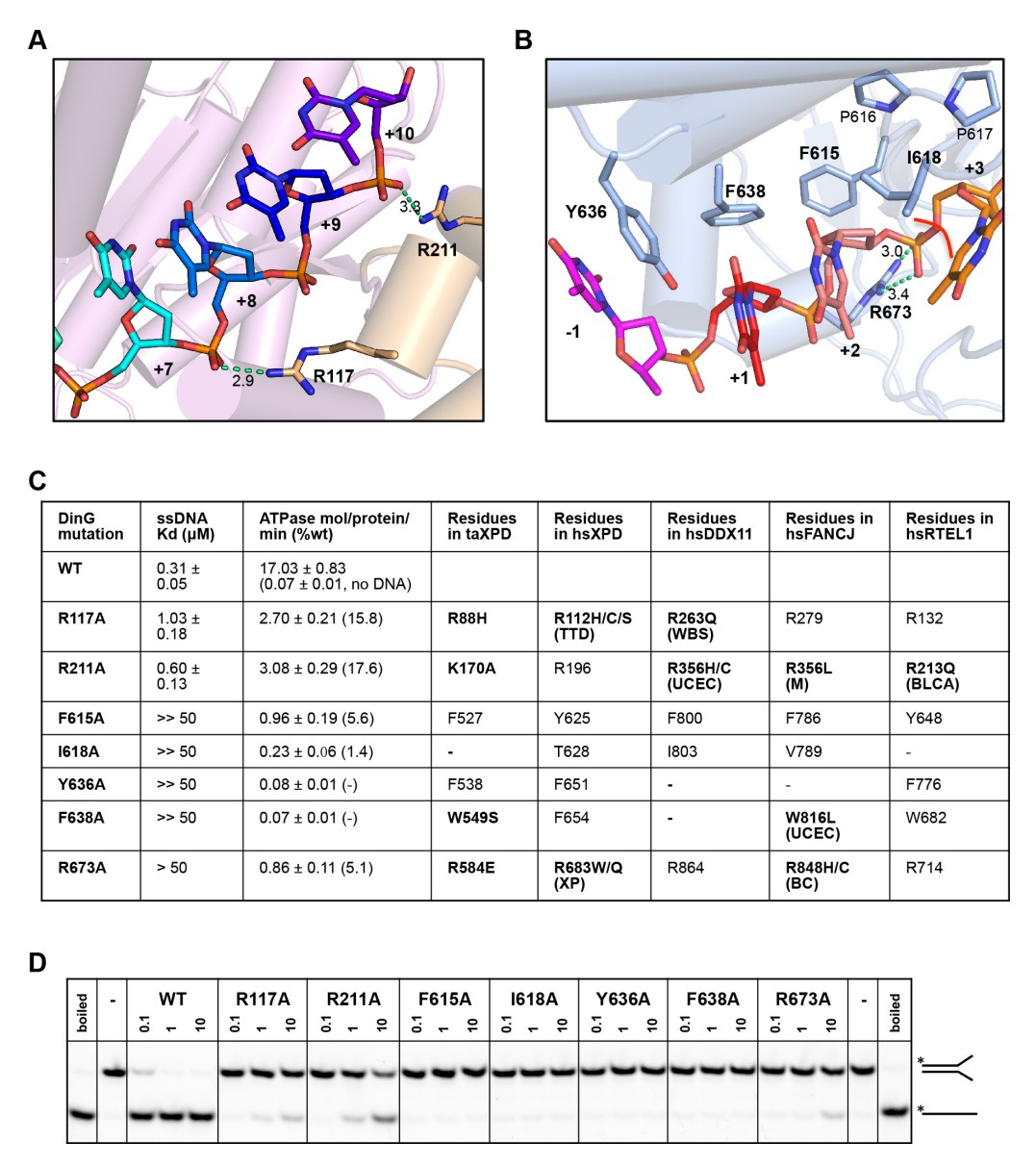

**Figure 4.** Biochemical data for mutant proteins. Residues contacting the 3'-end (**A**) or the 5'-end (**B**) of the bound ssDNA that were mutated. (**C**) Table showing ssDNA binding constants and ssDNA-dependent ATPase activity for wildtype DinG and mutants. The equivalent residues in *T. acidophilus* XPD (taXPD), human XPD (hsXPD), human DDX11 (hsDDX11), human FANCJ (hsFANCJ) and human RTEL1 (hsRTEL1) are listed and those highlighted in bold are directly linked to human disease. Residues in bold have been found to be mutated and confirmed important for protein function or diseased related (for human XPD, DDX11, FANCJ and RTEL1, the related disease is shown in brackets, 'TTD', Trichothiodystrophy, 'XP', Xeroderma Pigmentosum, 'WBS', Warsaw Breakage Syndrome, 'UCEC', Uterine Corpus Endometrial Carcinoma, 'M', Mixed cancer types, 'BC', Breast cancer, 'BLCA', Bladder Urothelial Carcinoma). This information was obtained from *Krassowski et al. (2018)* and *Cerami et al. (2012)*. (**D**) Comparison of helicase activities of wildtype and mutant DinG proteins. Forked DNA substrate (50 nM) was incubated at various concentrations (0.1, 1 and 10 μM) of enzyme and the products were analysed on 12% native-PAGE. Heat-denatured substrate was run as a positive control.
DOI: https://doi.org/10.7554/eLife.42400.016

The following figure supplement is available for figure 4:

**Figure supplement 1.** ssDNA-binding data for DinG mutants.
DOI: https://doi.org/10.7554/eLife.42400.017

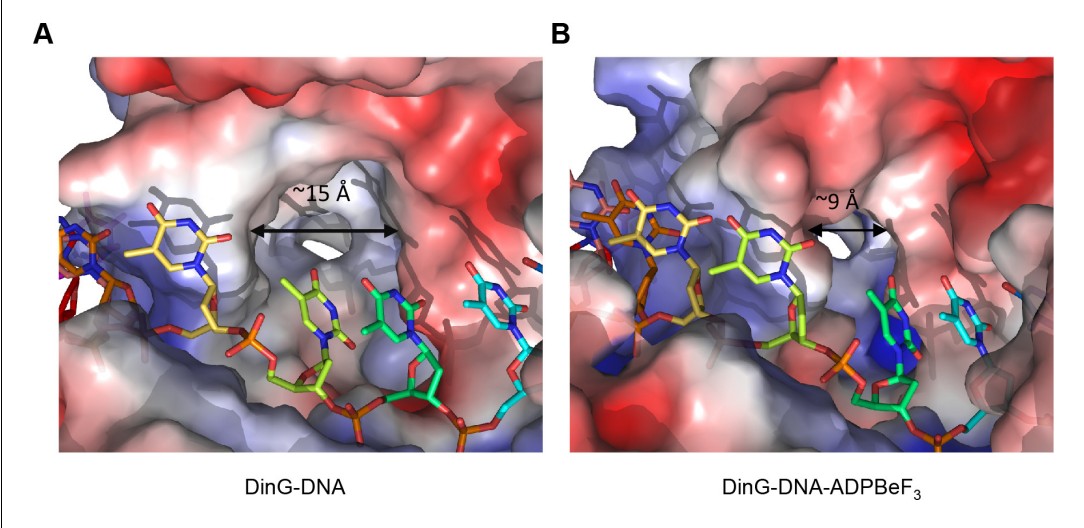

**Figure 5.** A pocket underneath the ssDNA binding site. The molecular surface of (**A**) the binary complex and (**B**) the ternary complex, coloured by electrostatic charge (negative in red, positive in blue). In the binary complex, a large hydrophobic pocket is present between the ATPase domains, beneath the ssDNA binding site that becomes restricted when ATP binds to the enzyme. The open pocket in the binary complex could accommodate much larger DNA lesions (e.g. cyclopyrimidine dimers) that could not be accommodated in the ternary complex, suggesting that binding of large lesions in this pocket would prevent ATP-induced conformational changes and thereby block translocation. Protein contact potential was calculated in PyMOL and shown as a surface with negative potential coloured in red and positive in blue. The ssDNA strand was omitted from the calculations but is overlaid as sticks.

DOI: https://doi.org/10.7554/eLife.42400.018

The following figure supplement is available for figure 5:

**Figure supplement 1.** A proposed mechanism for 5′−3′ translocation and potential DNA lesion checkpoints.

DOI: https://doi.org/10.7554/eLife.42400.019

# Materials and methods

## Key resources table

| Reagent type (species) or resource | Designation | Source or reference | Identifiers | Additional information |
| --- | --- | --- | --- | --- |
| Gene (*E. Coli, strain K12*) | DinG | NA | NCBI-GeneID: 945431 | |
| Strain, strain background (*E. coli BL21 (DE3)*) | *E. coli BL21 (DE3)* | NEB | C2527H | |
| Recombinant DNA reagent | Plasmid pET28-HMT -DinG (encoding full length *E.coli* DinG, fused with N-terminal 6 × His tag, MBP-tag and a TEV protease recognition site) | This study | | |
| Commercial assay or kit | QuikChangeTM Site-Directed Mutagenesis Kit | Stratagene, La Jolla, CA | La Jolla, CA | |
| Chemical compound, drug | PEG4000 | sigma | | |
| Chemical compound, drug | PEG3350 | sigma | | |
| Chemical compound, drug | BeCl3 | sigma | | |

*Continued on next page*

*Continued*

| Reagent type (species) or resource | Designation | Source or reference | Identifiers | Additional information |
|---|---|---|---|---|
| Chemical compound, drug | phosphoenolpyruvate | sigma | | |
| Chemical compound, drug | pyruvate kinase | sigma | | |
| Chemical compound, drug | lactate dehydrogenase | sigma | | |
| Chemical compound, drug | ATP | sigma | | |
| Chemical compound, drug | ADP | sigma | | |
| Software, algorithm | COOT | (*Emsley et al., 2010*) | https://www2.mrc-lmb.cam.ac.uk/personal/pemsley/coot/ | |
| Software, algorithm | Phenix | (*Adams et al., 2010*) | https://www.phenix-online.org/documentation/reference/real_space_refine.html | |
| Software, algorithm | Refmac | (*Murshudov et al., 1997*) | NA | |
| Software, algorithm | Xia2 | (*Winter et al., 2013*) | NA | |
| Software, algorithm | PyMOL | Schrodinger, LLC | https://pymol.org/2/ | |
| Software, algorithm | NT Analysis software (NanoTemper) | NA | https://nanotempertech.com/monolith/ | |

## Cloning and site-directed mutagenesis

A list of primers used for cloning and mutagenesis is provided in *Supplementary file 1*. The full-length gene encoding *E. coli* DinG (residues 1-716aa) was amplified from *E. coli* genomic DNA by PCR and cloned into a modified pET28a expression vector, pET28-HMT, which contains a fused N-terminal 6 × His tag, a MBP-tag and a TEV protease recognition site (His-MBP-TEV). Site directed mutagenesis was performed with a QuikChangeTM Site-Directed Mutagenesis Kit from Stratagene (La Jolla, CA) as described previously (*Cheng et al., 2016*). The fidelity of the mutants was confirmed by sequencing. Primers used for cloning and mutageneses are listed in *Supplementary file 1*.

## Protein preparation

In brief, transformed *E. coli* BL21 (DE3) clones were grown at 37°C in LB medium containing 50 µg/ml Kanamycin to an optical density at 600 nm of 0.6–0.8. Protein expression was induced at 30°C for 5 hr by adding isopropyl-β-D- thiogalactopyranoside (IPTG) to a final concentration of 0.8 mM. After harvesting, cells were re-suspended in lysis buffer (20 mM Tris (pH 8.0), 1 M NaCl, 0.5 mM TCEP and 5 mM imidazole), lysed by sonication and centrifuged at 18,000 × g for 60 min at 4°C. The supernatant was purified on a HisTrap HP column (GE Healthcare, Fairfield, CT), equilibrated with buffer A (20 mM Tris (pH 8.0), 1 M NaCl, 0.5 mM TCEP and 5 mM imidazole), washed with 30 mM imidazole and finally eluted with 300 mM imidazole. After TEV-tag-removal using TEV protease, the protein was dialysed into buffer B (20 mM Tris (pH 8.0), 250 mM NaCl and 0.5 mM TCEP) and re-loaded onto the HisTrap HP column (GE Healthcare) to remove the uncleaved protein. The flow-through fractions were collected and loaded onto a Heparin HP column (GE Healthcare) pre-equilibrated with buffer B. Fractions containing DinG protein was eluted with a linear gradient from 250 mM to 600 mM NaCl. The protein was finally purified by Superdex 200 10/300 GL column (GE Healthcare) with buffer C (20 mM Tris (pH 8.0), 300 mM NaCl and 0.5 mM TCEP). The mutant variants were expressed and purified in the same way. The concentration of proteins was estimated using the extinction coefficient of 78858 $M^{-1}$ $cm^{-1}$ at 278 nm, calculated from the amino acid sequence with the ProtParam tool (www.expasy.ch/tools/protparam.html).

## Crystallization and structure determination

Crystallization trials were carried out by the sitting drop vapour diffusion method at 293 K. Freshly purified DinG was concentrated to ~4 mg/ml and centrifuged to remove insoluble material before crystallization. To make the DinG-ssDNA complex, protein was mixed with 12 base oligo-dT in a 1:1.2 molar ratio. To prepare the ternary DinG-DNA- ADP•BeF$_3$ complex, 10 mM MgAc, 2 mM ADP, 5 mM NaF and 2 mM BeCl$_3$ were added sequentially and the mixture was incubated on ice for 10 min before setting up the crystallization. Crystals of DinG-ssDNA (binary complex) were obtained in 0.25 M KCl, 0.1M HEPES (pH 7.8) and 22% PEG 3350. Crystals of DinG-DNA-ADP•BeF$_3$ (ternary complex) were obtained in 0.1M MgCl$_2$, 0.1M HEPES (pH 7.0), and 15% PEG4000. Cryo-freezing was achieved as described previously (*Cheng et al., 2016*) before flash freezing in liquid nitrogen. X-ray diffraction data were collected on beamlines I04 and I04-1 at the Diamond synchrotron X-ray source, and were integrated and scaled with the Xia2 system (*Winter et al., 2013*). For selenium-labeled crystals, data were collected at two wavelengths, 0.9795 (peak) and 0.9860 (low remote).

Initial phases for the binary complex structure were calculated from the MAD data using PHENIX (*Adams et al., 2010*), and the MAD data analysis is shown in Table S2. The phases were applied to native data (*Table 1*) and this map was used for model building and refinement. The ternary complex structure was determined by molecular replacement using separated DinG domains from the binary model as search models in CCP4, followed by rigid-body refinement by Refmac5 (*Murshudov et al., 1997*). Structures were refined using PHENIX (*Adams et al., 2010*) and interspersed with manual model building using COOT (*Emsley et al., 2010*). Later stages of refinement utilized TLS group anisotropic B-factor refinement. The refined binary complex model contained one DinG molecule and one ssDNA molecule in the asymmetric unit, while the ternary complex model contained two DinG complexes in the asymmetric unit. All residues are in the most favourable and allowed regions of the Ramachandran plot. Structure figures were rendered in PyMOL. The statistics for data collection and model refinement are listed in *Table 1*.

## DNA binding assays

Microscale thermophoresis (MST) was used to determine ssDNA-binding constants for DinG or mutant variants (*Willhoft et al., 2016*). Interactions were measured on a temperature equilibrated Monolith NT.115 (NanoTemper), using a 5'FAM fluorescent labelled 14 base oligo-dT. For each set of measurements, a two fold serial-dilution of wild-type DinG or mutant variants concentration (starting at 80 µM) was prepared at 2x final in MST-Buffer (25 mM Tris (pH 8.0), 200 mM NaCl, 1 mM DTT, 5% Glycerol), before mixing with an equal volume of DNA substrate (also in MST-Buffer), to yield a final substrate concentration of 10 nM. Samples were equilibrated for 15 min before being loaded into Monolith NT.115 MST Premium Coated capillaries (NanoTemper). The system was given a further 15 min to re-equilibrate at 25°C before beginning each scan. The LED power was set at 20% and the MST power was set to 80% for all experiments. All other settings were left at the default values. Measurements were carried out in triplicates and with different protein batches. K$_d$ fitting was performed on the NT Analysis software (NanoTemper).

## Helicase assays

Substrate for helicase assays was created by annealing equal amounts of ssDNA 5'-TTTTTTTTTTTTTTAATGTATAATGCGAGCACTGCTACAGCACGG-3' and 5'- FAM-CCGTGCTGTAG-CAGTGCTCGCATTATACATTTTTTTTTTTTTTTT-3'. This forked DNA substrate contains 14 free bases on each tail, and 31 base pairs of duplex. Typically, 50 nM substrate was incubated with various concentrations (0.1, 1 and 10 µM) of DinG in a 10 µl reaction mixture containing 50 mM Tris (pH 8.0), 200 mM NaCl, 10 mM MgCl$_2$, 10 µg/ml bovine serum albumin, 1 mM DDT and 5% glycerol. Unwinding reactions were initiated by adding 2.5 mM ATP, followed by incubation at 37°C for 20 min and finally stopped by the addition of protease K, SDS and EDTA to 0.1 mg/ml, 1% and 10 mM, respectively. Each set of experiments contained a 'positive control' in which the helicase substrate was heat-denatured by incubation at 95°C for 5 min followed by quick chilling on ice. Products were separated in 12% TBE-native polyacrylamide gels. The gels were scanned and quantified in fluorescence mode (FAM) on a BIO-RAD ChemiDocTM MP imaging system.

## ATPase assays

ATPase activity of DinG was measured using a coupled assay that monitors ADP release as described previously (*Willhoft et al., 2016*). A final concentration of 100 µM NADH, 0.5 mM phosphoenolpyruvate, 100 U/ml pyruvate kinase (Sigma), 20 U/ml lactate dehydrogenase (Sigma) were used in all reactions in a final volume of 50 µl. Typical reactions were conducted using 20 nM purified DinG with or without 200 nM 14 base oligo-dT and 1 mM ATP in reaction buffer (50 mM Tris (pH 8.0), 100 mM NaCl, 10 mM $MgCl_2$, 10 µg/ml bovine serum albumin, 1 mM DDT). ATP solutions were made with a ratio of 1:1:2 ATP:$MgCl_2$:Tris base. Reactions were conducted by mixing all components immediately prior to transferring to an Optiplate-384 Black Opaque 384-well microplate (Perkin Elmer) that had been pre-incubated at 37°C. Reactions were initiated with the injection of ATP using the built-in reagent injectors, which were pre-filled with $10 \times$ ATP solution. Reactions were monitored fluorescently using an excitation of 335 nm and an emission of 469 nm at 37°C with a CLARIOstar microplate reader (BMG Labtech). The measurements were carried out in triplicate and with different protein batches. All reaction rates were determined from the initial linear rate and reaction kinetics were analysed assuming a Michaelis–Menten model.

## Acknowledgments

The work was funded by the Medical Research Council and Cancer Research UK (DBW). We thank Rhodri ML Morgan for assistance with data collection and Martin Wilkinson for advice during model building. We acknowledge the Diamond Light Source for access to beamlines I04 and I04-1. We acknowledge use of the crystallisation facility at Imperial College London, which is part-funded by the Wellcome Trust.

## Additional information

### Funding

| Funder | Author |
| --- | --- |
| Medical Research Council | Dale B Wigley |
| Cancer Research UK | Dale B Wigley |

The funders had no role in study design, data collection and interpretation, or the decision to submit the work for publication.

### Author contributions

Kaiying Cheng, Conceptualization, Investigation, Writing—original draft; Dale B Wigley, Conceptualization, Funding acquisition, Writing—original draft, Writing—review and editing

### Author ORCIDs

Dale B Wigley http://orcid.org/0000-0002-0786-6726

### Decision letter and Author response

Decision letter https://doi.org/10.7554/eLife.42400.027
Author response https://doi.org/10.7554/eLife.42400.028

## Additional files

### Supplementary files

• Supplementary file 1. Primers used for cloning and mutagenesis.
DOI: https://doi.org/10.7554/eLife.42400.020

• Transparent reporting form
DOI: https://doi.org/10.7554/eLife.42400.021

## Data availability

Diffraction data have been deposited in PDB under the accession codes 6FWR and 6FWS.

The following datasets were generated:

| Author(s) | Year | Dataset title | Dataset URL | Database and Identifier |
|---|---|---|---|---|
| Cheng K, Wigley DB | 2018 | Diffraction data from DNA translocation mechanism of an XPD family helicase | http://www.rcsb.org/structure/6FWR | Protein Data Bank, 6FWR |
| Cheng K, Wigley DB | 2018 | Diffraction data from DNA translocation mechanism of an XPD family helicase | http://www.rcsb.org/structure/6FWS | Protein Data Bank, 6FWS |

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
