## [Decision Letter]

Thank you for submitting your article "DNA translocation mechanism of an XPD family helicase" for consideration by *eLife*. Your article has been reviewed by two peer reviewers, and the evaluation has been overseen by a Reviewing Editor and John Kuriyan as the Senior Editor. The following individual involved in review of your submission has agreed to reveal his identity: Kevin Raney (Reviewer #1).

The reviewers have discussed the reviews with one another and the Reviewing Editor has drafted this decision to help you prepare a revised submission.

Summary:

The present work describes the structure of the DinG helicase, a member of the XPD family of helicases, bound to a ssDNA dT fragment (12 bases) in the absence and presence of ADP·BeF_3_. The structure provides important insights on how DinG and other members of the XPD family helicases translocate on ssDNA. Although the observations are similar overall to previous studies made by Wigley et al. suggesting a 1 base pair step size, the molecular details for movement of the ssDNA through the active site differs for DinG compared to other SF1 and 3'-to-5' SF2 helicases. The authors also present a model for how XPD family helicases may recognize bulky lesions based on their structure. The mechanistical impact of the data is significant for the field, as it represents a sub-family that has thus far eluded description in terms of translocation mechanism.

Essential revisions:

The reviewers have raised several questions that must be addressed in a revised manuscript. Only the first point, below, requires new experimental work.

1) The biochemical section should be completed with ATPase studies for the different variants. It is acknowledged that DNA binding defects can interfere with such an analysis; however, the Arg117 and Arg211 variants should be analyzed in particular, since there are also comparative data available for XPD. DDX11 is not discussed here despite its presence in the analysis.

2) The manuscript needs a more in-depth analysis of the data in the context of the current literature on iron sulfur cluster-containing helicases of the XPD family. In the Abstract two other family members, RTEL1 and FANCJ, are described as other prominent members of this family. However, DDX11 is not mentioned, yet is added in a table where DinG variants were analyzed.

The work points out that DinG is also involved in DNA repair in *E. coli*; however, the role of DinG in repair is not explained. Several works (e.g., Boubakri et al., 2012; Hwang et al., 2012; Grodick et al., 2014) should be considered as input for DinG function in DNA repair. It could then also be appreciated that DinG function in repair is not at all comparable to XPD or FANCJ. In addition, helicases like DDX11 or RTEL1 are not repair helicases per se that have a special function in a repair pathway as compared to XPD or FANCJ. It is important that a differentiated Introduction be provided that describes the major players concisely but sufficiently to appreciate the roles they assume in genome maintenance.

Some specific points along this theme include:

a) In the subsection “Overall structure of DinG and comparison with XPD”, the differences in the FeS domains of XPD and DinG should be described rather than just mentioning that they are different; a possible influence on DNA binding should also be at least considered. The two arginines in DinG could be more similar to the situation in MutY rather than XPD. Is there anything useful to be gained from considering MutY?

b) In the second paragraph of the subsection “Structure of the DinG binary complex with ssDNA”, it is mentioned that are differences between the structures of XPD and DinG, but these are not described nor are superpositions shown. The corresponding figure in the supplement in which XPD within TFIIH is superimposed with DinG is not very instructive, since it is not mentioned in the text at this position (Figure 1—figure supplement 2) and it does not explain what the significance of the differences is.

c) In the last paragraph of the subsection “Structure of the DinG binary complex with ssDNA”, differences between the FeS domains of XPD and DinG are again mentioned; however, it is difficult to discern in the sequence alignment where the residues are and how the differences manifest on a structural level. A detailed figure illustrating the comparative analysis, perhaps showing a superposition, should be implemented. The figure in the supplement is not very descriptive and the different perspective suggests that there is no significant homology between the FeS domains. Is this the case? If so, it raises questions about the relevance of the findings here for XPD and possibly for other family members as well. Similarly, in XPD, R88 also directly interacts with the FeS cluster, which does not seem to be the case for DinG. Can this be attributed to DNA binding or is this is a real structural difference? The answer may have implications for the generality of some of the findings presented here.

3) A more detailed comparative analysis of the current work with previous translocation mechanisms for other SF1 and SF2 helicases is needed. For example, in the last paragraph of the subsection “A mechanism for translocation of ssDNA in a 5’-3’ direction”, it is proposed that the DinG translocation mechanism is different to other mechanisms, these differences are not elaborated. Similarly, it is stated that the work here completes the mechanisms of translocation for SF1 and SF2 helicases in all directionalities; however, a detailed comparison of the data to previous data is lacking. As a consequence, a general reader cannot appreciate these differences nor the uniqueness of the findings.

4) Many of the mutants discussed in Figure 4 are disease related and should be more thoroughly explained. It should be delineated in which disease specific mutations are implicated and the disease should be explained briefly but correctly. Not all of the mutations mentioned are implicated in cancer as stated in the subsection “Biochemical data in support of the model”, (R112 for instance causes TTD). Warsaw breakage syndrome (DDX11) is not associated with cancerous malignancies. It is important that the basis and possible outcome of every variant analyzed for the respective protein be discussed and explained in the correct context.

5) Regarding damage recognition, has it been shown that all family members of the XPD family come across bulky lesions in their respective pathways? For example, the thymidine dimer mentioned is a very specific nucleotide excision repair (NER) substrate. Since there is no mention of NER in the manuscript, what are the implications here? DDX11, FANCJ and RTEL1 likely do not come across such a lesion in vivo. In prokaryotes, the UvrABC system takes care of NER, so the likelihood that DinG might verify such a lesion is questionable. There are no biochemical data testing a role for the pocket in damage sensing; if there likewise is no sequence or structural conservation of pocket residues within XPD, then this speculation should probably be omitted.

References:

Grodick MA, Segal HM, Zwang TJ, Barton JK. DNA-Mediated Signaling by Proteins with 4Fe–4S Clusters Is Necessary for Genomic Integrity. J. Am. Chem. Soc., 2014, 136 (17), pp 6470–6478. DOI: 10.1021/ja501973c

Hwang J, Lee K, Phadtare S, Inouye M. Identification of Two DNA Helicases UvrD and DinG as Suppressors for Lethality Caused by Mutant cspA mRNAs. J Mol Microbiol Biotechnol, 2012, 22:135–146. DOI: 10.1159/000339832

---

## [Author Response]

Essential revisions:The reviewers have raised several questions that must be addressed in a revised manuscript. Only the first point, below, requires new experimental work.1) The biochemical section should be completed with ATPase studies for the different variants. It is acknowledged that DNA binding defects can interfere with such an analysis; however, the Arg117 and Arg211 variants should be analyzed in particular, since there are also comparative data available for XPD. DDX11 is not discussed here despite its presence in the analysis.

As requested, we have added the ATPase activity data, notably for R117A and R211A (Figure 4) to the manuscript although we still feel their value is limited given the inability to saturate with ssDNA in most cases. We have corrected the disease references in the figure legend for all of the proteins involved including DDX1.

2) The manuscript needs a more in-depth analysis of the data in the context of the current literature on iron sulfur cluster-containing helicases of the XPD family. In the Abstract two other family members, RTEL1 and FANCJ, are described as other prominent members of this family. However, DDX11 is not mentioned, yet is added in a table where DinG variants were analyzed.

We have added more information about DDX11 in the Abstract and Introduction of the revised manuscript.

The work points out that DinG is also involved in DNA repair in E. coli; however, the role of DinG in repair is not explained. Several works (e.g., Boubakri et al., 2012; Hwang et al., 2012; Grodick et al., 2014) should be considered as input for DinG function in DNA repair. It could then also be appreciated that DinG function in repair is not at all comparable to XPD or FANCJ. In addition, helicases like DDX11 or RTEL1 are not repair helicases per se that have a special function in a repair pathway as compared to XPD or FANCJ. It is important that a differentiated Introduction be provided that describes the major players concisely but sufficiently to appreciate the roles they assume in genome maintenance.

Unfortunately, the precise role of DinG in bacteria is unclear other than a general involvement in DNA repair as judged by mainly genetic studies. Indeed, the name of the gene (DinG) is DNA damage Inducible as it was identified in a genetic screen for DNA damage repair along with several other “Din” genes (e.g. DinB – a bypass polymerase). Some recent studies do indicate a role in R-loop removal and we have now added these to the Introduction. However, we note that many helicases have multiple roles in cells (e.g. UvrD) that can have specific roles in DNA damage repair pathways with other proteins of that pathway as well as roles in replication fork repair or transcription. It may well be that DinG also has multiple roles in cells, but that remains unclear at present. We hope our revised Introduction makes that more explicit now.

Some specific points along this theme include:a) In the subsection “Overall structure of DinG and comparison with XPD”, the differences in the FeS domains of XPD and DinG should be described rather than just mentioning that they are different; a possible influence on DNA binding should also be at least considered. The two arginines in DinG could be more similar to the situation in MutY rather than XPD. Is there anything useful to be gained from considering MutY?

We now provide additional details of the comparison between different FeS domains both in the revised figure (Figure 1—figure supplement 3) and in the revised manuscript. The main purpose of the comparison is the interaction with basic residues that occupy similar positions relative to the bound ssDNA despite coming from different parts of the FeS domains.

Since the structure of MutY is quite different to these enzymes, and the interaction is with dsDNA rather than ssDNA, we do not feel a comparison is useful.

b) In the second paragraph of the subsection “Structure of the DinG binary complex with ssDNA”, it is mentioned that are differences between the structures of XPD and DinG, but these are not described nor are superpositions shown. The corresponding figure in the supplement in which XPD within TFIIH is superimposed with DinG is not very instructive, since it is not mentioned in the text at this position (Figure 1—figure supplement 2) and it does not explain what the significance of the differences is.

In the second paragraph of the subsection “Structure of the DinG binary complex with ssDNA”, we are comparing the differences between previously published XPD-DNA complex structures (Kuper et al., 2012; Constantinescu-Aruxandei et al., 2016) which are not consistent with one another. No reference is made to Figure 1—figure supplement 2 at this point. However, we have added an additional panel to Figure 2—figure supplement 3 to show the differences between the two XPD-DNA complex structures (now panels B and C).

c) In the last paragraph of the subsection “Structure of the DinG binary complex with ssDNA”, differences between the FeS domains of XPD and DinG are again mentioned; however, it is difficult to discern in the sequence alignment where the residues are and how the differences manifest on a structural level. A detailed figure illustrating the comparative analysis, perhaps showing a superposition, should be implemented. The figure in the supplement is not very descriptive and the different perspective suggests that there is no significant homology between the FeS domains. Is this the case? If so, it raises questions about the relevance of the findings here for XPD and possibly for other family members as well. Similarly, in XPD, R88 also directly interacts with the FeS cluster, which does not seem to be the case for DinG. Can this be attributed to DNA binding or is this is a real structural difference? The answer may have implications for the generality of some of the findings presented here.

We have revised Figure 1—figure supplement 3 to compare the FeS domains of DinG and XPD. We superimpose structures of the FeS domains of other XPD family members (*Thermoplasma acidophilum* XPD, PDB ID: 4A15; *Saccharomyces cerevisiae* Rad3 (XPD), PDB ID: 5OQM; *Sulfolobus acidocaldarius* XPD, PDB ID: 3CRV; *Homo sapiens* XPD, PDB ID: 5OF4; and DinG in this study). The point we are trying to make here is that although the structures of the FeS domains differ, the basic residues that contact the bound ssDNA in DinG are conserved in space and these residues are conserved between other XPD family members that do have more similar FeS domains. This suggests they may play a similar role in interacting with ssDNA. The mutational data in Figure 4 confirm the role in binding ssDNA in DinG. These residues are of unknown function in XPD, RTEL1, FANCJ and DDX11 yet crop up as disease causing mutations in these proteins. In XPD, mutations at these residues affect DNA binding but by an unknown manner. Our structural model of DinG now suggests a potential role for these residues in other XPD family members which is why we felt this was a useful addition to the analysis.

Although true that R88 in taXPD interacts directly with the FeS cluser, that is not the case for other XPD structures. The bound ssDNA substrate in the taXPD complex is too short to reach to R88 so the interaction with the FeS cluster observed here (which is not seen in other XPD structures) may have little relevance.

3) A more detailed comparative analysis of the current work with previous translocation mechanisms for other SF1 and SF2 helicases is needed. For example, in the last paragraph of the subsection “A mechanism for translocation of ssDNA in a 5’-3’ direction”, it is proposed that the DinG translocation mechanism is different to other mechanisms, these differences are not elaborated. Similarly, it is stated that the work here completes the mechanisms of translocation for SF1 and SF2 helicases in all directionalities; however, a detailed comparison of the data to previous data is lacking. As a consequence, a general reader cannot appreciate these differences nor the uniqueness of the findings.

To address this point, we provide a new figure (Figure 3—figure supplement 1) to compare the translocation mechanisms for SF1 and SF2 helicases. Details of each mechanism would be much too complicated to present but the figure legend addresses the major differences between each mode of translocation.

4) Many of the mutants discussed in Figure 4 are disease related and should be more thoroughly explained. It should be delineated in which disease specific mutations are implicated and the disease should be explained briefly but correctly. Not all of the mutations mentioned are implicated in cancer as stated in the subsection “Biochemical data in support of the model”, (R112 for instance causes TTD). Warsaw breakage syndrome (DDX11) is not associated with cancerous malignancies. It is important that the basis and possible outcome of every variant analyzed for the respective protein be discussed and explained in the correct context.

As requested, we have corrected the references to the correct disease types for the point mutations in the table (Figure 4C) with references to appropriate databases for further information.

5) Regarding damage recognition, has it been shown that all family members of the XPD family come across bulky lesions in their respective pathways? For example, the thymidine dimer mentioned is a very specific nucleotide excision repair (NER) substrate. Since there is no mention of NER in the manuscript, what are the implications here? DDX11, FANCJ and RTEL1 likely do not come across such a lesion in vivo. In prokaryotes, the UvrABC system takes care of NER, so the likelihood that DinG might verify such a lesion is questionable. There are no biochemical data testing a role for the pocket in damage sensing; if there likewise is no sequence or structural conservation of pocket residues within XPD, then this speculation should probably be omitted.

We have been careful not to suggest that DinG plays the same role in bacteria as XPD in eukaryotes, nor indeed the same functions as other XPD family members (RTEL1, DDX11 and FANCJ) which likely differ from one another. However, the similarity between the structures of XPD (the only other family member for which there is currently structural information) suggests a commonality across all family members in the mechanism of 5’-3’ translocation on ssDNA, with potentially an additional similarity to some family members (e.g. XPD) in an ability to recognise bulky DNA lesions. The downstream repair pathways may be, and indeed are almost certainly, quite different in bacteria and eukaryotes and between different systems. Indeed, as the reviewers point out, the roles of different family members (c.f. XPD and RTEL1) are likely quite different even within eukaryotes. We have pointed out the potential for a mechanism of bulky DNA lesion recognition and response by XPD and DinG. However, this may not be required by other family members, and for clarity we have now stated that in the Introduction. By contrast, the mechanism for ssDNA translocation in a 5’-3’ direction, which is the main point of this study, is likely to be conserved.